# Mesenchymal Stem Cells in Homeostasis and Systemic Diseases: Hypothesis, Evidences, and Therapeutic Opportunities

**DOI:** 10.3390/ijms20153738

**Published:** 2019-07-31

**Authors:** Francisco J. Vizoso, Noemi Eiro, Luis Costa, Paloma Esparza, Mariana Landin, Patricia Diaz-Rodriguez, Jose Schneider, Roman Perez-Fernandez

**Affiliations:** 1Research Unit, Fundación Hospital de Jove, Avda. Eduardo Castro, 161, 33290 Gijón, Spain; 2Department of Pharmacology, Pharmacy and Pharmaceutical Technology, Faculty of Pharmacy, University of Santiago de Compostela-Campus Vida, 15782 Santiago de Compostela, Spain; 3Department of Obstetrics and Gynecology, Universidad Rey Juan Carlos, Avda. de Atenas s/n, 28922 Alcorcón, Spain; 4Department of Physiology-Center for Research in Molecular Medicine and Chronic Diseases (CIMUS), University of Santiago de Compostela, 15706 Santiago de Compostela, Spain

**Keywords:** Regenerative medicine, aging diseases, diabetes, lupus, secretome, conditioned medium, extracellular vesicles, exosomes

## Abstract

Mesenchymal stem cells (MSCs) are present in all organs and tissues, playing a well-known function in tissue regeneration. However, there is also evidence indicating a broader role of MSCs in tissue homeostasis. In vivo studies have shown MSC paracrine mechanisms displaying proliferative, immunoregulatory, anti-oxidative, or angiogenic activity. In addition, recent studies also demonstrate that depletion and/or dysfunction of MSCs are associated with several systemic diseases, such as lupus, diabetes, psoriasis, and rheumatoid arthritis, as well as with aging and frailty syndrome. In this review, we hypothesize about the role of MSCs as keepers of tissue homeostasis as well as modulators in a variety of inflammatory and degenerative systemic diseases. This scenario opens the possibility for the use of secretome-derived products from MSCs as new therapeutic agents in order to restore tissue homeostasis, instead of the classical paradigm “one disease, one drug”.

## 1. Introduction

Many chronic diseases of inflammatory and/or degenerative origin do not currently have satisfactory treatment. The advent of regenerative medicine based on stem cells could provide new promising alternatives. Among the various stem cell types, i.e., hematopoietic, embryonic, induced pluripotent, and mesenchymal, the mesenchymal stem cells (MSCs) are awakening the most extraordinary interest [1] due to the absence of serious adverse effects reported following MSC transplantation, unlike those associated with the allogeneic transplant of hematopoietic stem cells [2].

MSCs were first described in the 1950s by the Russian haematologist A. Friedenstein [3]. They are a heterogeneous group of multipotent cells, morphologically akin to fibroblasts, that form colonies and are capable of differentiate into mesenchymal lineages [4,5]. Although in small amounts, MSCs have been isolated from numerous organs and tissues, such as bone marrow, adipose tissue, umbilical cord, dermis, muscle, synovial membrane, peripheral blood, tonsil, periodontal ligament, dental pulp and uterus, among others [6,7] (some of them summarized in Table 1 and Table 2), suggesting a perivascular origin since perivascular cells natively express MSC markers [8]. However, MSCs subtypes differ in their biological features [9,10].

To refer to mesenchymal-like cells various nomenclatures are used as “mesenchymal stem cells”, “mesenchymal stromal cells” and “multipotent stromal cells”, but the acronym MSCs is now generally used to identify this class of cells. Because of the initial variation in nomenclature and characterization, the International Society for Cellular Therapy established the minimum criteria required for MSCs definition as follows: (a) plastic-adherent cells when maintained in standard culture conditions; (b) expression of CD105, CD73 and CD90, and lack of expression of CD45, CD34, CD14 or CD19, CD79a or CD11b, and HLA-DR surface molecules, and (c) capacity to differentiate into adipocytes, osteoblasts, and chondroblasts in vitro [11].

Many studies have demonstrated that secretome-derived products from MSCs, such as exosomes and conditioned medium, have therapeutic effects on key pathological processes that are associated with basic homeostatic functions, such as cell differentiation and proliferation, angiogenesis and vasculogenesis, inflammation, and oxidative stress (Table 1 and Table 2).

In addition, recent studies have shown the capacity of MSCs to exert antimicrobial effects, indicating an immune function independent of the host’s immune system [44]. Therefore, this experimental and clinical evidence strongly suggests the physiological relevance of MSCs in tissue homeostasis. Because of these properties, MSCs are currently being used in Phase I and II clinical trials in several pathologies, including immunological, bone, heart or neurodegenerative disorders [45], and even in phase III clinical trials in graft-versus-host disease (GVHD), Crohn’s disease, myocardial infarction and liver cirrhosis [1].

This present review addresses aspects of MSCs, such as mechanisms of intercellular communication, their dysfunction in different physio-pathological processes, their role in homeostasis, and their possible therapeutic use.

## 2. MSCs and Its Secretome in Intercellular Communication

Several studies have demonstrated that intravenously injected MSCs can migrate specifically to the sites of tissue damage, such as those caused by ischemic conditions or inflammation [46]. Even, it has been demonstrated that systemic administration of MSC was more efficient at all-time points for engraftment compared to after local MSC transplantation [47]. In addition, unlike other stem-cell-based therapies, MSCs do not require differentiation into a mature cell type prior to administration and have strong homing capacities in the damaged sites after cell transplantation [48]. However, the molecular mechanism underlying the efficacy of MSCs in promoting engraftment and the functional recovery of injury sites is still unclear [49]. Studies of the potential of MSCs to treat cardiovascular diseases, have shown the ability of MSCs to form new blood vessels by differentiating into endothelial cells in vivo [50,51]. However, other studies have shown poor viability and survival of transplanted cells into the host tissue [52,53,54] and often less than 1% of transplanted MSCs are long-term retained within the target tissue [55,56].

This suggests that the beneficial effects of their transplantation are not the result of the cells themselves, but rather are related to their ability to secrete bioactive factors which provide a favorable microenvironment to injured tissues and help limit the damage area and promote regenerative response [57,58]. In fact, MSC-derived products can effectively mimic the therapeutic effects of MSCs in preclinical models. These secreted bioactive factors may generically be termed “secretome or conditioned medium”. This biological product includes molecular soluble factors such as cytokines and growth factors, but also membrane-bound vesicles containing biomolecules. As shown in Table 1, Table 2 and Table 3, some of these factors are involved in homeostatic and therapeutic actions at multiple levels. As it shown in Table 1 and Table 2, probably the most plausible scientific evidence of the biological effects of the MCS-secretome derived products are their reported actions after local administration in several experimental in vivo models. Either the whole conditioned medium or the extracellular vesicles (EVs) obtained from different human origin MSC cultures perform extensive therapeutic benefits. EVs are particles made up of phospholipid membranes that contain growth factors, cytokines, lipids, DNA and various forms of RNA. They represent an intercellular communication pathway and play an important role in several cellular mechanisms, such as the exchange of genetic material, the transfer of biologically active molecules, and the defense against viral attack in mammalian cells [59]. Indeed, EVs interact with recipient cells by mechanisms which resemble those involved in viral entry [60]. Although exosomes are secreted by almost all human cell types, protective effects seem specific to MSC-derived exosomes, unlike, for example, fibroblast-derived exosomes [61]. MSCs secretome may contain three different types of EVs: (a) Exosomes (40–150 nm in diameter); (b) microparticles (50–1000 nm in diameter), and (c) apoptotic bodies (500–2000 nm in diameter).

## 3. MSCs Dysfunction in Systemic Diseases and Aging

The idea of MSC dysfunction in systemic diseases arose from the observation that patients with autoimmune diseases such as systemic lupus erythematosus (SLE), diabetes mellitus (DM), rheumatoid arthritis (RA), and multiple sclerosis entered disease remission when treated with mesenchymal or hematopoietic stem cells after allogenic transplants, but not after autologous transplants. These findings were especially relevant in patients with lymphoma or leukemia and a concomitant autoimmune disease, such as psoriasis [71]. It has been hypothesized that the remission took place due to the “resetting” of immune memory, but it could also be due to the restoration of internal homeostasis by the administration of external well-functioning MSCs.

Several recent studies indicate an altered functioning of MSCs in various systemic diseases, the role of MSCs in their pathogenesis and/or the development of associated comorbidities. These alterations may be acquired. One example is the systemic autoimmune disease RA, characterized by cartilage and bone destruction associated with local production of inflammatory mediators. Some studies have demonstrated that increased local production of TNFα may injure the bone marrow (BM) microenvironment and affect the reserves of BM haematopoietic progenitor cells [72]. Moreover, a significant reduction in MSC expansion through passages has been observed in patients with RA suggesting a defective proliferative capacity [73]. Dysfunction in MSCs from several sources (e.g., bone marrow, adipose tissue, umbilical cord, and dermis) has been associated with a number of diseases (Table 4). Among these diseases, SLE and DM are representative of the possible impact on their systemic pathophysiology of MSC dysfunction.

SLE is a chronic inflammatory disease that affects all major organs and systems of the body. Inflammation has long been proposed as a cause for accelerated aging. Early studies reported that BMSCs from lupus patients, compared to matched controls, had a flattened morphology, proliferated more slowly, showed increased ROS, had increased expression of p16^INK4a^ and increased activation of the p^53^/^p21^ pathway [74]. In addition, genetic alterations have been shown in SLE which have a direct or indirect role in MSC immune-regulation function [112]. For instance, the OAZ transcription factor is over-expressed in MSCs from SLE patients, which impairs MSC regulation of B cells, leading to anti-nuclear antibody production [113]. Similarly, p16^ink4a^, an inhibitor of cyclin-dependent kinase CDK4 and CDK6, related to senescence of MSCs [114], shows increased expression in MSCs from SLE patients, inhibiting TGF-β secretion and contributing to the decrease of Treg cells [81].

DM is the most common metabolic disease. Over 382 million people (8.3% of the world population) are affected, with an estimated increase to 592 million in the next 20 years [115]. DM leads to many life-threatening complications affecting major organs, such as heart, kidneys, and eyes [116]. It has been reported MSCs to adopt an insulin-secreting phenotype [117,118]. In association with DM, this hyperglycaemic state is considered a stressor that leads to a pathological microenvironment and compromises MSC functionality. Four types of MSC anomalies are found in DM: altered pro-inflammatory cytokine secretion, altered cellular differentiation and proliferation, changes in angiogenesis/vasculogenesis, and increased oxidative stress. Firstly, in the diabetic milieu, inflammatory cytokines, such as IL-6, are chronically elevated [119]. Considering that MSCs function is highly regulated by cytokines [96,98,120], this might constitute a relevant aspect of the disease. In fact, changes in cytokine interactions can induce altered patterns in the MSC secretome [121], which mediates critical cell signaling and migratory pathways [68]. Secondly, regarding differentiation and proliferation of MSCs, many of the complications that arise in diabetes could be the result of MSC dysfunction [97]. MSCs have also been shown to display an increased tendency to differentiate into adipocytes in diabetic states, which may contribute to the disease burden [95,97,122]. In addition, the increased tendency of diabetic MSCs to differentiate into adipocytes is often coupled with reduced differentiation into osteoblasts, which has been suggested as the cause of the increased bone fractures and osteoporosis in diabetic patients [123,124]. Thirdly, the delicate balance of factors implicated in angiogenesis is well-known to be altered in the diabetic state [94]. Several studies have reported the impaired angiogenic capacity of MSCs as a result of different alterations, such as decreased expression of major angiogenic genes (e.g., VEGF-A, VEGF-C, angiopoietin 1 and angiopoietin 2) [125] or decreased expression of proteins required for endothelial migration and vascular smooth muscle formation (e.g., VE-cadherin and α-SMA) [92]. In addition, one subpopulation of MSCs, specifically associated with an elevated angiogenic and vasculogenic gene profile, is expressed at a lower proportion in type 1 and 2-DM cell populations compared to controls [122]. And fourthly, oxidative stress and autophagy has been related the MSC dysfunction in patients with metabolic syndrome and type 2 DM [126].

With respect to aging, by definition stem cells cannot be fully senescent. Their inability to undergo permanent cell cycle arrest is precisely what defines their ability to divide and repopulate. However, MSC functionality declines with aging. In fact, MSCs in early passages have shown better colony efficiency than in later passages [127], which should be taken into consideration for therapeutic purposes.

Several MSC senescence phenotypes have been recognized, such as an increase in flattened morphology, growth arrest in G1 phase of cell cycle, increased expression of senescence-associated lysosomal α-l-fucosidase and senescence-associated β-galactosidase [128]. In addition, reparative capacity of MSCs may decrease with age [129], and MSCs obtained from aged individuals possess reduced immunomodulatory properties compared to those from younger ones [130]. MSCs from both bone marrow and adipose tissues present reduced capacity to handle oxidative stress with increasing donor age [131]. Oxidative stress leads to hyperactivity of pro-growth pathways, such as insulin/IGF-1 and mTOR, and the subsequent accumulation of toxic aggregates and cellular debris ultimately leading to apoptosis, necrosis, or autophagy [132].

MSC senescence may be involved in the loss of tissue homeostasis, which could lead to organs failure and development of age-related diseases. In this sense, there are MSC alterations associated with their multilineage differentiation, homing, immunomodulatory and wound-healing capacity, oxidative stress regulation and intrinsic changes in telomere shortening [133,134,135,136]. Collectively, these aging-related stem cells changes ultimately lead to Frailty Syndrome [137]. Frailty has been clinically defined as “a state of increased vulnerability resulting from aging-associated decline in reserve and function across multiple organ systems, such that the ability to cope with every day or acute stressors is compromised” [138]. This may be because of the full senescence of stem cells, and is considered as stem cell exhaustion. Regenerative medicine has been proposed to offer further therapeutic approaches to improve or reverse frailty signs and symptoms [135]. In fact, deterioration of adult stem cells in the adult phase can become an important player in the onset of several aging diseases, such as the metabolic syndrome [139], diabetes mellitus [140,141], rheumatoid arthritis [73], systemic lupus erythematosus [80] or ageing syndromes [142,143]. These diseases are characterized by the perpetuation of inflammatory states, constant emission of “alarm signals,” proliferation, mobilization, and finally an endless sequestration of MSCs into the damaged tissues, probably leading to a decrease in the endogenous pool of MSCs, which are perhaps the most important specialized repairing cells [144]. This could lead to irreversible and premature stem cell exhaustion syndrome (SCES), inhibiting the organism to self-repair and survive.

Aging-related diseases are highly relevant. According to the 2018 Aging and Health report of the World Health Organization, 2 billion people worldwide (22% of the population) will be over the age of 60 by the year 2050, which is more than double the rate in 2015 (http://www.who.int/mediacentre/factsheets/fs404/en/). Therefore, a new era of therapeutics focusing on the restoration of MSC functionality could be promising.

## 4. Control of Tissue Homeostasis by MSCs: Hypothesis and Therapeutic Opportunities

Based on the above, the following scenario regarding tissue homeostasis by MSCs leads us to an interesting hypothesis. A damaged somatic cell might send “alarm signals” indicative of dysfunction in the form of exosomes, for example. These membrane-derived vesicles could then be internalized by “resident sentinel” MSCs and would trigger their proliferation and activation in response to the damage in the somatic cell, leading ultimately to the production of a specialized secretome. This secretome, able to establish an intercellular communication and with regenerative, anti-inflammatory, and other above-described properties, would be decisive in restoring the physiological balance in the damaged cell, and by extension, in the whole organ (Figure 1A). This intercellular communication might involve, among others, the newly discovered microanatomical fluid-filled space within and between tissues [145].

According to this hypothesis, a number of different situations could lead to loss of tissue regulation control: (a) inadequate alarm messages by damaged somatic cells (Figure 1B); (b) inadequate response to those alarm signals by MSCs due to their depletion (Figure 1C); (c) inadequate response to alarm signals by MSCs due to primary or secondary cell dysfunction, induced by alterations in tissue microenvironment (Figure 1D); and (d) inadequate response by somatic cells to the intercellular communication signals coming from MSCs (Figure 1E).

This new physio-pathological paradigm, together with possible variants, could offer new opportunities in the diagnosis and treatment of homeostasis dysfunction through the identification of quantitative or qualitative alterations corresponding to each of the involved stress signals and the response by MSCs. In a therapeutic setting, the impaired equilibrium could be theoretically restored using a specifically designed cocktail of substances aimed at palliating the dysfunction in intercellular communication between damaged somatic cells and MSCs.

For this purpose, secretomes derived from cultured MSCs, including EVs, could be potential candidates, as these secreted and/or derived products have demonstrated their potential for repairing organs and tissues damaged by various degenerative and/or inflammatory disorders (Table 1 and Table 2) [6,146]. Furthermore, secretome-derived EVs have therapeutic advantages as they have the ability to protect their cargo from unfavorable environmental conditions, such as changes in pH or digestive (lytic) enzymes into the bloodstream and damaged tissues.

## 5. MSC-Derived Secretome Products as Therapeutic Agents

The use of MSC-derived secretome products offers key advantages over applications based on stem cells themselves [6]. These advantages include: greater safety, by avoiding issues associated with transplantation of living and proliferative cell populations; better evaluation of MSC-derived secretome regarding dosage and potency, such as conventional pharmaceutical agents; better storage (without presence of potentially toxic cryopreservative agents for a long period and without loss of potency); economical mass-production through tailor-made cell lines under controlled laboratory conditions; and the possibility of being immediately available for acute disease treatment. In addition, MSC-derived secretome could be modified for more effective therapeutic applications.

Nevertheless, several important related aspects must be borne in mind when envisaging further applications of MSCs-derived secretome and its derivates, such as MSC origin, donor condition (age, sex, and health status), as well as several technical and biological aspects related to the development of secretome-derived products

### 5.1. Origin of MSCs

Proteomic comparison of MSC-derived secretomes from different tissue sources have revealed differing profiles and capabilities. For example, MSC-derived secretome from bone marrow, adipose tissue, and dental pulp present different protein composition [147]. It has been also reported that Wharton’s jelly-derived MSCs secrete greater amounts of proinflammatory proteins and growth factors, while those derived from adipose tissue have an enhanced angiogenic profile and secrete greater amounts of extracellular matrix proteins and metalloproteases [148]. Thus, the origin of MSCs seems an important aspect related to their possible therapeutic uses. In fact, a MSC population obtained from the human uterine cervical transformation zone [69], displays age-related properties which may affect the regression rate of cervical intraepithelial neoplasia by means of paracrine effects [149]. It has also been recently shown that the conditioned medium from those cells has growth-inhibiting properties against different microorganism species of *Candida*, a common pathogen of the vaginal medium, to which the cervical transformation zone is in permanent contact [70].

### 5.2. Donor Condition

Theoretically identical MSC populations from different individuals may display different secretome properties, depending on factors including age and health status [150]. Thus, for example, as mentioned above, MSCs obtained from aged individuals possess reduced immunomodulatory properties compared to those from younger subjects, and MSCs from patients affected by several diseases exhibit reduced capabilities [130,131]. These points should be considered in the development and application of secretome-derived products, using specific functional tests to ensure homogeneity of action.

### 5.3. Bioprocess Development for Secretome-Derived Products

There are several aspects related to the technical development of secretome-derived products which may influence their potency, such as the specific platform on which cells are grown and culture conditions.

Commonly, MSCs are grown in 2D monolayers in tissue culture flasks (T-flasks). However, this labor-intensive methodology involves a large number of T-flasks with the risk of flask-to-flask variability and contamination [151]. An alternative for mass production of MSCs is the use of bioreactors in which cells grow homogeneously in 3D suspension [152]. This highly scalable technology allows the cells to form three-dimensional aggregates (spheroids), which are considered to be more biomimetic and capable of increasing the levels of reparative/regenerative, anti-inflammatory and angiogenic factors [153].

Among the culture conditions which may influence the quality of MSC-derived secretome products are: type of media and supplements (e.g., fetal bovine serum, xeno-free, or chemically-defined media), temperature, pH, seeded-cell density, oxygen level, and mechanical, electromagnetic, or biochemical stimuli (e.g., lipopolysaccharide (LPS), TNF-α, TNF-β, INF-γ or hydrogen peroxide -H_2_O_2_-) [154]. For example, in response to hypoxia, MSCs increase the production of several angiogenic and anti-apoptotic factors, such as VEGF, IL-6, CCL2, and stanniocalcin-1 (STC-1) [155,156].

These data suggest that it may be possible to adapt secretome-derived products to individual patients (Figure 2). However, it is still necessary to get insight into large-scale production of MSC-derived secretome according to the Good Manufacturing Practices (GMP) guidelines.

Secretome-derived products can be understood as a combination of therapeutic biomacromolecules and vesicles, and thus treated as biopharmaceutics for the development of delivery platforms. The design of delivery systems must guarantee stability, allow easy administration and maximize pharmacological effects. The nature of secretome-based products should allow their loading on micro/nanoparticulated systems of variable composition and structure, which have been widely studied for biomacromolecular therapeutics. This type of carriers can increase macromolecules half-life in vivo, control drug delivery profiles and allow for specific targeting reducing side effects [157,158,159]. Moreover, the use of other polymeric-based nanostructured drug delivery platforms such as hydrogels has also been widely used for the delivery of biopharmaceuticals. They present similarities to the native extracellular matrix in terms of oxygen and nutrient permeability while showing excellent biocompatibility and porous structure [160,161]. Moreover, the use of injectable hydrogels allows for their administration in the target sites with minimal invasiveness permitting local biomacromolecule retention and delivery [162].

In summary, there are a wide variety of drug delivery strategies that could be tested for secretome-derived product administration to maintain/increase their potency and efficiency. Secretome-derived products loaded on delivery platforms, could open new possibilities of restoring tissue homeostasis in a controlled manner in time and/or place.

## 6. Conclusions and Future Perspectives

A growing body of evidence suggests the significant role for MSCs in the regulation of tissue homeostasis. These cells are widely distributed throughout the human body and considered the most important type of stem cells involved in tissue regeneration. The positive regenerative, immunoregulatory, proangiogenic, antitumor, and antimicrobial activity of MSCs has also been demonstrated in their secretome-derived products in several in vivo experimental models.

Recently, knowledge has increased regarding the role of morphological and functional alterations of MSCs in several important systemic diseases and the aging process. A paradigm shift could result if the hypothesis is confirmed that alterations in intercellular communication signals between somatic cells and MSCs are key in the occurrence of diseases. This would entail the development of new therapeutic strategies based on the recognition of damaged signals and their restoration, instead of the classical paradigm of “one disease, one drug”. This context would require the standardization of secretome-derived products, their manufacture and individual adaptation to each pathological process and patient.

## Figures and Tables

**Figure 1 ijms-20-03738-f001:**
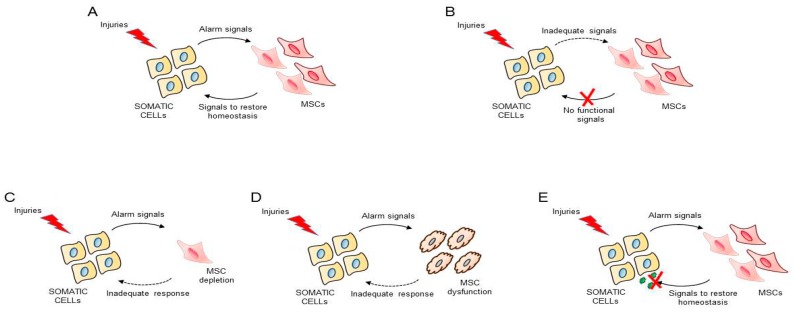
Possible intercellular communication stages between somatic cells and MSCs: (**A**) damaged somatic cells might send “alarm signals” indicative of dysfunction to “resident sentinel” MSCs which would trigger their proliferation and activation in response to the damage in the somatic cell, leading ultimately to the production of a specialized secretome; (**B**) inadequate alarm messages by damaged somatic cells; (**C**) inadequate response to those alarm signals by MSCs due to their depletion; (**D**) inadequate response to alarm signals by MSCs due to primary or secondary cell dysfunction, induced by alterations in tissue microenvironment; and (**E**) inadequate response by somatic cells to the intercellular communication signals coming from MSCs.

**Figure 2 ijms-20-03738-f002:**
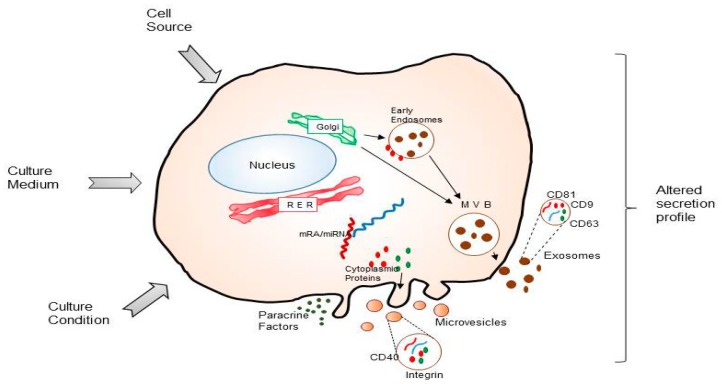
Factors related to bioprocess development (cell source, culture medium and culture conditions) which may influence the quality of MSC secretome-derived products (paracrine factors, microvesicles or exosomes).

**Table 1 ijms-20-03738-t001:** Therapeutic effects of mesenchymal stem cell (MSC)-derived exosomes on disease in vivo models.

Disease	MSC Source	Administration Via	Experimental Model	Therapeutic Effect	Ref
**Local administration**					
**Diabetic wound healing**	Gingival	Topical	Diabetic rat	Promotion of healing in diabetic skin defects.	[12]
	Synovium	Topical	Diabetic rat	Stimulation of proliferation of human dermal fibroblasts and human microvascular endothelial cells.	[13]
**Corneal epithelial wound**	Corneal	Local	Mouse	Acceleration of corneal epithelial wound healing.	[14]
**Traumatic and degenerative ocular disease**	Bone marrow	Intravitreal injection	Rat	Promotion of retinal ganglion cells and axon regeneration survival.	[15]
**Autistic-like behaviors**	Bone marrow	Intranasal	BTBR mice	Increase of male to male social interaction and reduce repetitive behaviors.	[16]
**Liver fibrosis**	Umbilical cord	Intra-hepatic	Mouse	Decrease of surface fibrous capsules and alleviate hepatic inflammation.	[17]
**Periodontitis**	Adipose-derived	Local Injection	Rat	Increase in newly organized tissue.	[18]
**Systemic administration**					
**Cutaneous wound healing**	Adipose tissue	Intravenous	Mouse	Acceleration of cutaneous wound healing and stimulation of fibroblast migration and collagen synthesis.	[19]
	Umbilical cord	Subcutaneous injection	Rat	Promotion of wound healing and angiogenesis.	[20]
	Adipose tissue	Intravenous injection	Mouse	Promotion of extracellular matrix reconstruction and regulation of fibroblast differentiation to mitigate scar formation.	[21]
	Menstrual blood-derived	Intradermic injection	Mouse	Resolution of inflammation, reepithelization accelerated by induction of M1-M2 macrophage polarization and increased neoangiogenesis.	[22]
**Atopic dermatitis**	Adipose tissue	Intravenous and subcutaneous injection	Mouse	Decrease of clinical score, level of serum IgE, number of eosinophils in blood and infiltration of mast cells, CD86+ and CD206+ cells. Decrease of mRNA expression of pro-inflammatory cytokines.	[23]
**Hepatic injury**	Umbilical cord	Intravenous	Mouse	Reduction of oxidative stress and apoptosis.	[24]
**Endotoxin-induced acute lung injury**	Bone marrow	Intravenous	Mouse	Reduction of white blood cells and neutrophils from bronchoalveolar lavage fluid (BALF).	[25]
**Bronchopulmo-nary dysplasia**	Wharton jelly Bone marrow	Intravenous	Mouse	Amelioration of alveolar simplification, fibrosis and pulmonary vascular remodelling, reduction of pro-inflammatory M1, and increase of anti-inflammatory M2 macrophages.	[26]
**Osteonecrosis**	Synovial membrane	Intramuscular	Rat	Prevention of osteonecrosis, enhance proliferation and anti-apoptotic effects.	[27]
**Local and systemic administration**					
**Pneumonia/*E. coli***	Bone marrow	Intratracheal Intravenous	Mouse	Reduction of lung injury, white blood cells and neutrophils in BALF. Reduction of *E. coli* in BALF, lung and blood. Increased survival.	[28]
**Lung injury**	Bone marrow	Intratracheal Intravenous	Mouse	Reduction of lung injury, white blood cells, neutrophils, total protein, MIP-1 and *E. coli* in BALF. Increase of survival.	[29]
	Wharton jelly	Intratracheal	Mouse	Reduction of lung edema, airway resistance, pulmonary artery pressure, neutrophils in lung, and inflammatory cytokines in BALF. Increase of KGF, PGE2 and IL-10 in BALF.	[30]
**Lung fibrosis/Silica**	Bone marrow	Intratracheal	Mouse	Reduction of calcified nodules size, hydroproline in lung, and inflammatory cells in BALF.	[31]
	Bone marrow	Intratravenous	Mouse	Reduction of lung collagen and white blood cells in BALF.	[32]

**Table 2 ijms-20-03738-t002:** Therapeutic effects of MSC-derived conditioned medium on disease in vivo model.

Disease	MSC Source	Administration Via	Experimental Model	Therapeutic Effect	Ref
**Local administration**					
**Cutaneous wound healing**	Bone marrow	Local	T1 diabetic rats	Acceleration of wound healing.	[33]
**Keloid**	Adipose tissue	Local	Mouse	Inhibition of proliferation and collagen synthesis of human keloid-derived fibroblast. Reduction of inflammation and fibrosis.	[34]
**Dry eye and corneal epithelial wound**	Uterine cervix	Local	Rat	Improvement in wound healing of alkali-injured corneas. Strong bactericidal effect on infected corneal contact lens	[35]
			Rabbit	Improvement in epithelial regeneration Reduction of corneal pro-inflammatory cytokines.	[36]
**Uveitis**	Uterine cervix	Topical	Mouse	Reduction of inflammation, and LPS-induced pro-inflammatory cytokines. Decrease in leucocytes in aqueous humor and ocular tissues.	[37]
**Systemic administration**					
**Acute liver failure**	Bone marrow	Intravenous	Rat	Inhibition of liver injury biomarkers release and promotion of recovery in liver structure.	[38]
**Multiple sclerosis**	Periodontal ligament	Intravenous	Mouse	Decrease in clinical and histologic score, and modulation of inflammation, oxidative stress, and apoptotic pathways.	[39]
**Diabetes**	Adipose tissue	Intravenous	Mouse	Reverse mechanical, thermal allodynia and thermal hyperalgesia. Restoration of pro/anti-inflammatory cytokine balance. Prevention of skin innervation loss and re-establishment of Th1/Th2 balance. Recovery of kidney morphology.	[40]
**Pneumonia/*E. coli***	Bone marrow	Intravenous	Rat	Increase in survival.	[41]
**Acute kidney injury**	Bone marrow	Intramuscular	Rat	Amelioration of kidney injury.	[42]
**Myocardial infarct**	Bone marrow	Intravenous and intracoronary	Porcine	Reduction of myocardial infarct size. Improvement of systolic and diastolic cardiac performance.	[43]

**Table 3 ijms-20-03738-t003:** Bioactive factors in MSC-derived secretome.

Bioactive Effects	Factors	Ref
Proliferation/Regeneration	FGFs, HGF, IGF-1, EGF, PDGF, VEGF, TIMP-1, TIMP-2, UPAR	[35,62,63,64]
Angiogenesis	FGFs, HGF, IGF-1, IL-6, MCP-1, PDGF, VEGF	[35,62,64]
Anti-apoptosis	FGF, IL-6, IGF-1, GM-CSF, HGF	[35,62,63,64,65]
Anti-fibrosis	FGFs, HGF, TIMP-1, MMPs	[35,62,64,66]
Chemo-attraction	CCLs, CXCLs, G-CSF, LIF, MCP-1	[35,62,65,67]
Immuno-modulation	IDO, IL-10, IL-6, LIF, NT-3, PGE-2	[37,62,67,68]
Anti-tumoral	FLT-3, CXC10/IP10, LAP, Light	[69]
Bactericidal	CXC10/IP10, CXCL8/IL8, CXCL1/GRO-7, CXCL6/GCP-2, CCL20/MIP-3, CCL5/RANTES	[35,62]
Antifungal	IL-6, IL-8, IL-17, IP-10, CCL-5, CXC-6, CXC-16	[70]

**Table 4 ijms-20-03738-t004:** MSC dysfunction in diseases.

Disease	MSC Source	MSC Features	Ref
		Flattened morphology.	[74,75,76]
		Increased cell senescence and apoptosis.	[77]
		Impaired potential for differentiation and migration.	[78]
**Systemic** **Lupus** **Erythematosus**	Bone marrow	Increased activation of the p^53/p21^ pathway.	[79,80]
		Increased expression of p16^INK4a^	[80,81]
		Increased reactive oxygen species.	[80]
		Alteration of expression profiles in genes related to immune function.	[74,80,82,83,84]
**Idiopathic pulmonary fibrosis**	Bone marrow	Mitochondrial dysfunction, with accumulation of DNA damage. Cell senescence. Decreased capacity to migrate. Increased pro-inflammatory responses.	[85]
		Impaired differentiation and decreased proliferation.	[86,87,88,89]
**Diabetes** **mellitus**	Bone marrow and Adipose tissue	Impaired angiogenesis/vasculogenesis.	[90,91,92,93,94,95]
		Increased pro-inflammatory cytokines.	[96]
		Greater propensity to differentiate into adipocytes.	[97]
	Umbilical cord	Increased pro-inflammatory cytokines.	[98]
		Reduced ex vivo proliferation and clonogenic potential, premature senescence, and accelerated shortening of telomere terminal restriction fragments.	[99]
**Multiple sclerosis**	Bone marrow	Reduced in vitro neuroprotective potential.	[100]
		Reduced expression, activity, and secretion of key antioxidants. Increased susceptibility to nitrosative stress.	[101]
**Rheumatoid arthritis**	Bone marrow	Impaired proliferative potential in association with premature telomere length loss.	[73]
**Parkinson disease**	Bone marrow	Impaired differentiation, mitochondrial dysfunction and increased ROS generation and oxidative stress.	[102]
**Amyotrophic lateral sclerosis**	Bone marrow	Reduced migration.	[103]
		Alterations in metalloproteases.	[104]
		Reduced capacity of pluripotency and trophic factor secretion.	[103,105]
**Psoriasis**	MSCs in psoriasis plaques or from areas surrounding the psoriasic eruptions	Increased expression of inflammation and angiogenesis-related genes.	[106,107,108,109,110]
**Myelodysplastic syndromes**	Bone marrow	Altered morphology, reduced proliferative potential, p53 pathway activation, dysregulated miRNA in extracellular vesicles.	[111]

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
