# Peer review of "Mesenchymal Stem Cells in Homeostasis and Systemic Diseases: Hypothesis, Evidences, and Therapeutic Opportunities"

_ijms, 2019, doi:10.3390/ijms20153738_

Round 1
Reviewer 1 Report
Now, the manuscript was revised satisfactory.
Reviewer 2 Report
This review has been significantly altered since the initial submission which I already evaluated high enough for publication in IJMS.
Review of this type would certainly be a beneficial publication for Journal reputation and interesting to a Reader.
In its recent form it provides a summarised overview of MSC-based cell therapy including historical milestones and recent advances.
This work is a good contribution to the field and the Journal's reputation
This manuscript is a resubmission of an earlier submission. The following is a list of the peer review reports and author responses from that submission.
Round 1
Reviewer 1 Report
A huge number of research papers have been published about the Mesenchymal stem cell transplantation for the past 30 years, as well as a large number of review papers have also been published. Thus, in my honest opinion, the theme of this review article is not particular and not become intrigued. Therefore, the drastic cutting edge may be a point of publication. However, it is likely that the contents are also not particular and much more common.
Therefore, my major comments are as follows.
1) First of all, I think it is desperately needed to the usage of the word “Mesenchymal stem cells (MSC)”. Actually, this is across-the-broad and confusing word. In particular, the description “tissue specific MSC” is the most confusing. Basically, MSC should be limitedly used to “bone marrow MSC” following the nature of this word (referred as ref. 1), and the other tissue specific stem cells should be described more clearly, such as “adipose tissue-derived stem cells” and/or “Wharton jelly-derived stem cells”. In this context, description that “MSCs have been isolated and characterized in numerous organs and tissues” is confusing, whereas the sentence was followed by “suggesting a perivascular origin”. Thus, I recommend that the authors should be made the new section, such as “the definition of MSC in this review” and clearly state that there are summarized in the Table 1, for example. Then, the authors also should state that other tissue specific stem cells are eliminated in this review, because a large number of the other tissue specific stem cells have been identified and reported during recent two decades, but there are missed in this review. This is helpful for the readers confine to the conception and/or work origin against the description “many studies have demonstrated”, because the origin of “many studies” was summarized in Table 1. Researchers whose adopting other tissue specific stem cells feel odd in the current style.
2) Through the MSCs transplantation studies, the biggest problem is a lack and/or less about evidence of cell engraftment and/or migration into the therapeutic reference tissues. Thus, yes or no to the verification of cell engraftment and/or topical migration should be included in Table 1. Without this evidence, it is hard to discuss the therapeutic effects of MSC-derived exosomes (no cells no exosomes). For example, therapeutic effect to the myocardial infarction is particularly doubtful, and similar kind of questions likely observed other organ/tissue studies. The papers reported with no mechanical/clear evidences, but as if effective, should be eliminated from the references in order to prevent confusion. The papers described “how effective” with clear and direct evidences should be collected and reviewed. This may be a cutting-edge observation.
Author Response
Reviewer 1
1) First of all, I think it is desperately needed to the usage of the word “Mesenchymal stem cells (MSC)”. Actually, this is across-the-broad and confusing word. In particular, the description “tissue specific MSC” is the most confusing. Basically, MSC should be limitedly used to “bone marrow MSC” following the nature of this word (referred as ref. 1), and the other tissue specific stem cells should be described more clearly, such as “adipose tissue-derived stem cells” and/or “Wharton jelly-derived stem cells”. In this context, description that “MSCs have been isolated and characterized in numerous organs and tissues” is confusing, whereas the sentence was followed by “suggesting a perivascular origin”. Thus, I recommend that the authors should be made the new section, such as “the definition of MSC in this review” and clearly state that there are summarized in the Table 1, for example. Then, the authors also should state that other tissue specific stem cells are eliminated in this review, because a large number of the other tissue specific stem cells have been identified and reported during recent two decades, but there are missed in this review. This is helpful for the readers confine to the conception and/or work origin against the description “many studies have demonstrated”, because the origin of “many studies” was summarized in Table 1. Researchers whose adopting other tissue specific stem cells feel odd in the current style.
As suggested by the Reviewer, we have clarified the different type of stem cells (page 1, lines 32-33): “Among the various stem cell types, i.e. hematopoietic, embryonic, induced pluripotent, and mesenchymal, the mesenchymal stem cells (MSCs), (…)”. Also, we have corrected the sentence of line 39: “MSCs have been isolated from numerous organs and tissues, (…)” and explain the concept of perivascular origin (line 40): “(…), suggesting a perivascular origin since perivascular cells natively express MSC markers.” As recommended by the Reviewer, we have explained the nomenclature used (lines 43-45) “To refer to mensenchymal-like cells various nomenclatures are used as “mesenchymal stem cells”, “mesenchymal stromal cells” and “multipotent stromal cells”, but the acronym MSCs is now generally used to identify this class of cells.” In addition, we have described other tissue specific stem cells and we have stated that the source of MSCs considered in the review are summarized in the table 1 and 2 (lines 39-42): “(…), such as bone marrow, adipose tissue, umbilical cord, dermis, muscle, synovial membrane, peripheral blood, tonsil, periodontal ligament, dental pulp and uterus, among others (some of them summarized in Table 1 and 2).”
2) Through the MSCs transplantation studies, the biggest problem is a lack and/or less about evidence of cell engraftment and/or migration into the therapeutic reference tissues. Thus, yes or no to the verification of cell engraftment and/or topical migration should be included in Table 1. Without this evidence, it is hard to discuss the therapeutic effects of MSC-derived exosomes (no cells no exosomes). For example, therapeutic effect to the myocardial infarction is particularly doubtful, and similar kind of questions likely observed other organ/tissue studies. The papers reported with no mechanical/clear evidences, but as if effective, should be eliminated from the references in order to prevent confusion. The papers described “how effective” with clear and direct evidences should be collected and reviewed. This may be a cutting-edge observation.
The concept of the present manuscript is to review the paracrine effect of MSCs, therefore the cell engrafment of cell migration into the reference tissues is not considered. Regarding the Reviewer’s comment “no cells no exosomes”, please refer to page 5 lines 78-79 “Either the whole conditioned medium or the extracellular vesicles (EVs) obtained from different human origin MSC cultures perform extensive therapeutic benefits.” With the aim to clarify this concept we have corrected and clearly stated the following sentence (page 2, lines 49-50): “Many studies have demonstrated that secretome-derived products from MSCs, such as exosomes and conditioned medium, have therapeutic effects on key pathological processes (…)(Table 1 and 2).”

Reviewer 2 Report
Dear colleagues!
It is a comprehensive work and a well-assembled review, which I had a pleasure to read as an assigned Reviewer.
The review focuses on new aspects of cell therapy and reasonably reconsiders previously obtained data from respected groups and trials. Authors cite sufficient number of high-impact works and concluding part provides clear direction for development of the field. Review of this type would certainly be a beneficial publication for Journal reputation and interesting to a Reader
Author Response
Reviewer 2
It is a comprehensive work and a well-assembled review, which I had a pleasure to read as an assigned Reviewer.
The review focuses on new aspects of cell therapy and reasonably reconsiders previously obtained data from respected groups and trials. Authors cite sufficient number of high-impact works and concluding part provides clear direction for development of the field. Review of this type would certainly be a beneficial publication for Journal reputation and interesting to a Reader.
We want to thank the Reviewer for its kind comment.

Round 2
Reviewer 1 Report
I totally unconvinced the authors' response; The concept of the present manuscript is to review the paracrine effect of MSCs, therefore the cell engraftment of cell migration into the reference tissues is not considered. Analysis of cell exosomes in the culture medium is no problem, because there are cells in culture dishes and their exosomes absolutely including in the medium, this is quite general and reasonable way to detect paracrine exosomes in vitro. However, in the case of in vivo, there are no proofs of cell engraftment/migration, but paracrine effect exerted, is unreasonable. Thus, “no cells no exosomes”. If the transplanted MSCs lives in the blood streams (no proofs probably), how long living, and how delivered exosomes to the reference tissues should be clarified.
About the Table 3, there are mostly in vitro proofs; is there an in vivo proof after transplantation? Therapeutic effects, which had no direct proof should be indicated and eliminated is my point, to be establish the really stem cell therapy. Sorting the in vivo transplantation reports, how effective with and without direct evidence, is there any problem? For example, therapeutic effect; amelioration of kidney injury; but no direct evidence of engraftment/migration and/or with the proof of migration into kidney cortex etc. I think no problem, and this is a cutting-edge observation/review.